# Environmental Burden of Childhood Disease in Europe

**DOI:** 10.3390/ijerph16061084

**Published:** 2019-03-26

**Authors:** David Rojas-Rueda, Martine Vrijheid, Oliver Robinson, Aasvang Gunn Marit, Regina Gražulevičienė, Remy Slama, Mark Nieuwenhuijsen

**Affiliations:** 1Barcelona Institute of Global Health (ISGlobal), 08003 Barcelona, Spain; martine.vrijheid@isglobal.org (M.V.); mark.nieuwenhuijsen@isglobal.org (M.N.); 2Municipal Institute of Medical Research (IMIM-Hospital del Mar), 08003 Barcelona, Spain; 3Universitat Pompeu Fabra (UPF), Departament de Ciències Experimentals i de la Salut, 08003 Barcelona, Spain; 4CIBER Epidemiología y Salud Pública (CIBERESP), 28029 Madrid, Spain; 5Environmental and Radiological Health Sciences, Colorado State University, Fort Collins, CO 80523, USA; 6MRC-PHE Centre for Environment and Health, School of Public Health, Imperial College London, London SW7 2AZ, UK; o.robinson@imperial.ac.uk; 7Department of Air Pollution and Noise, Division for Infection Control and Environmental Health, Norwegian Institute of Public Health, N-0213 Oslo, Norway; Gunn.Marit.Aasvang@fhi.no; 8Department of Environmental Sciences, Faculty of Natural Sciences, Vytauto Didžiojo Universitetas, 44248 Kaunas, Lithuania; r.grazuleviciene@gmf.vdu.lt; 9Department of Prevention and Therapy of Chronic Diseases, Institute of Advanced Biosciences (IAB), Inserm—CNRS—University Grenoble—Alpes, 38700 Grenoble, France; remy.slama@univ-grenoble-alpes.fr

**Keywords:** environmental health, burden of disease, disability-adjusted life years, childhood, Europe

## Abstract

*Background*: Environmental factors determine children’s health. Quantifying the health impacts related to environmental hazards for children is essential to prioritize interventions to improve health in Europe. *Objective*: This study aimed to assess the burden of childhood disease due to environmental risks across the European Union. *Methods*: We conducted an environmental burden of childhood disease assessment in the 28 countries of the EU (EU28) for seven environmental risk factors (particulate matter less than 10 micrometer of diameter (PM_10_) and less than 2.5 micrometer of diameter (PM_2.5_), ozone, secondhand smoke, dampness, lead, and formaldehyde). The primary outcome was disability-adjusted life years (DALYs), assessed from exposure data provided by the World Health Organization, Global Burden of Disease project, scientific literature, and epidemiological risk estimates. *Results*: The seven studied environmental risk factors for children in the EU28 were responsible for around 211,000 DALYs annually. Particulate matter (PM_10_ and PM_2.5_) was the main environmental risk factor, producing 59% of total DALYs (125,000 DALYs), followed by secondhand smoke with 20% of all DALYs (42,500 DALYs), ozone 11% (24,000 DALYs), dampness 6% (13,000 DALYs), lead 3% (6200 DALYs), and formaldehyde 0.2% (423 DALYs). *Conclusions*: Environmental exposures included in this study were estimated to produce 211,000 DALYs each year in children in the EU28, representing 2.6% of all DALYs in children. Among the included environmental risk factors, air pollution (particulate matter and ozone) was estimated to produce the highest burden of disease in children in Europe, half of which was due to the effects of PM_10_ on infant mortality. Effective policies to reduce environmental pollutants across Europe are needed.

## 1. Introduction

Childhood is considered an important stage of life because children are more vulnerable than adults to many environmental risk factors [1]. This vulnerability results from the biological sensitivity that is an inherent characteristic of early growth and development [1]. Environmental exposures have been proposed as important health determinants for children and adults [2]. In particular, environmental exposures such as air pollution, heavy metals, and secondhand smoke, among others, have been suggested as important contributors to the environmental disease burden in all ages [3]. So far, only a few studies have estimated the environmental burden of disease during childhood [1,4,5,6,7]. These studies have focused on a specific group of environmental pollutants [7] or based their estimations on expert judgment (using qualitative approaches, without including a comparative risk assessment) and not on robust evidence (using studies other than large cohort studies or meta-analysis) [1,4,7].

In this study we aimed to estimate the burden of childhood disease due to environmental risk factors in the European Union (EU) of the 28 countries, describing the impact of seven environmental exposures (particulate matter less than 10 micrometer of diameter (PM_10_) and less than 2.5 micrometer of diameter (PM_2.5_), ozone, secondhand smoke, dampness, lead, and formaldehyde), identifying priorities in environmental health policies for childhood in Europe, and highlighting research and risk management necessities. These seven environmental risk factors were chosen, from many possible environmental risk factors for children, based on the strength of evidence on the causal relationship between the risk factor and a health outcome, and the data (populational, health, and exposure) availability across the EU of 28 countries. The environmental risk factors were defined for this study as a physical or chemical environmental exposure that harms the health of the children. The scope of this study does not include non-environmental risk factors, such as lifestyle, metabolic, genetic, or those related to access to health services. This document is divided into five sections (introduction, methods, results, discussion, and conclusion). The methods section describes the exposures and outcome selection, data collection, and burden of disease methods. The results section summarizes the impacts on the EU of 28 counties and describes the results for each exposure and the sensitivity analysis. The discussion section compares the results with previous studies and explains the differences between exposures and geographical distributions and includes a list of recommendations for authorities, public health practitioners, and researchers.

## 2. Methods

### 2.1. Selection of Environmental Risks and Health Outcomes

This burden of disease focused only on environmental risk factors in children. Metabolic and behavioral risk factors (e.g., sedentary, nutrition, active smoking), genetics, and those related to access to health services were excluded from the assessment. The selection of environmental risks and health outcomes was based on the following criteria: evidence for a causal relationship between exposure to the environmental risk factor and the health effect (based on meta-analyses, World Health Organization (WHO) guidelines, or previous risk assessments), independent health effects between the risk factors, availability of exposure data at national level, and availability of baseline health statistics at national level. Exposure-response functions and references used are shown in Table 1.

### 2.2. Data Collection

Our study considered the population between 0 to 18 years old of the 28 European Union Countries (EU28) (Austria, Belgium, Bulgaria, Croatia, Cyprus, the Czech Republic, Denmark, Estonia, Finland, France, Germany, Greece, Hungary, Ireland, Italy, Latvia, Lithuania, Luxemburg, Malta, Netherlands, Poland, Portugal, Romania, Slovakia, Slovenia, Spain, Sweden, and the United Kingdom). Population data by country and age were collected from the Eurostat database for the year 2015 [15] and the Institute for Health Metrics and Evaluation (IHME) [16].

Health data was collected from the Institute for Health Metrics and Evaluation (IHME) [16,17] and the World Health Organization [18,19] for asthma, mild mental retardation, otitis media, lower respiratory tract infections, and infant mortality. Data for cough and low respiratory tract symptoms [20] were collected from scientific papers and reports. Exposure data were collected from the IHME [17] for lead, PM_10_, PM_2.5_, ozone, and secondhand smoke, the Environmental and Health Information System (ENHIS) [21] for dampness, and from previous studies for formaldehyde [3] (see Appendix A). Lead exposure data from the IHME [17] database was primarily extracted from the literature regarding blood lead concentrations. Blood lead values were derived from studies that take blood samples and analyze them using various techniques to determine the level of lead present. The theoretical minimum-risk exposure level (TMREL) for lead used was 2.0 mg/dL. For PM_10_ and PM_2.5_, exposure estimates came from the IHME [17] and were drawn from multiple sources, including satellite observations of aerosols in the atmosphere, ground measurements, chemical transport model simulations, population estimates, and land-use data. Monitor-specific measurements rather than city averages was used. The measurements were recorded in 2014. For locations measuring only PM_10_, PM_2.5_ measurements were estimated from PM_10_. This was performed using a locally derived conversion factor (PM_2.5_/PM_10_ ratio for stations where measurements are available for the same year), which was estimated using population-weighted averages of location-specific conversion factors for the country. Satellite estimates were available at 11 × 11 km resolution and combine aerosol optical depth retrievals from multiple satellites with the GEOS-Chem chemical transport model and land use information. Estimates of the sum of particulate sulfate, nitrate, ammonium, and organic carbon and the compositional concentrations of mineral dust simulated using the GEOS-Chem chemical transport model, and a measure combining elevation and the distance to the nearest urban land surface were available for 2000 to 2015 for each 11 × 11 km grid cell. The TMREL for PM_10_ was assumed to be 7.9 μg/m^3^ and uniform distribution from 2.4–5.9 μg/m^3^ of PM_2.5_. For ozone, exposure data were derived from the TM5-FASST chemical transport model, which generates a 3-month running average of daily 1-hour maximum ozone values at the 0.1° × 0.1° for the years 1990, 2000, and 2010 [17]. The TMREL of ozone was also defined based on the exposure distribution from the American Cancer Society CPSII study. A uniform distribution was drawn around the minimum and 5th percentile values experienced by the cohort. This value was 33.3–41.9 ppb. Secondhand smoke exposure was estimated based on the Health Behavior in School-aged Children survey collected between 2013 and 2014. The TMREL for secondhand smoke was zero exposure. For dampness, exposure levels were derived from the Eurostat Statistics for Income and Living Conditions (SILC) data [17]. The TMREL for dampness was zero exposure. For formaldehyde, country levels were obtained from national indoor concentration data reported in the scientific literature [3]. For those countries without national concentration reporting, the lowest level reported in other European countries was used. The TMREL level used for formaldehyde was 100 mg/m^3^.

### 2.3. Environmental Burden of Disease

The environmental burden of disease analysis was performed following the comparative risk assessment approach proposed by the World Health Organization [22] and the global burden of disease project [2]. The environmental burden of childhood disease was estimated for the year 2015, using population data from 2015 and exposure date ranging from 2013 to 2015. Exposures thresholds, if any, were based on a counterfactual exposure distribution that would result in the lowest population risk. The feasibility of reaching the counterfactual exposure levels in practice was not assessed here. The burden of disease was estimated using the exposure data and a relative risk (RR) derived from epidemiological studies to estimate the population attributable fraction (PAF). This analysis was applied to each exposure-outcome pair. The PAF is defined as the proportional reduction in disease or death that would occur if exposure to the risk factor were reduced to the counterfactual. The burden of disease was calculated using disability weights (DW) and estimates of duration (L) of each outcome.

See below the explanation using three general formulas:
(1)PAF = f × (RR − 1)/f × (RR − 1) + 1,(2)AP = PAF × P,(3)DALY = AI × DW × L,
where PAF: population attributable fraction; f: fraction of population exposed; RR: relative risk; AP: attributable prevalence; P: background prevalence; DALY: disability-adjusted life year; DW: disability weights; and L: duration of condition.

The disability weights used in this analysis were those proposed by the global burden of disease project [23] or used in previous environmental burden of disease estimations [3]. For some cases an approach based on the unit of risk (UR) was used. The UR estimated the absolute number of cases that are to be expected at a certain exposure and then was transformed to DALYs using disability weights and the duration of the condition. Burden from lead and mental retardation, PM_2.5_ and low respiratory infections, secondhand smoke and low respiratory infections, and otitis media were obtained from the global burden of disease project 2015 [2].

## 3. Results

### 3.1. Environmental Burden of Disease in the EU of 28 Countries

Seven different exposures were identified under the inclusion criteria, associated to six different health outcomes (Table 1 and Table 2). We estimated that in the population aged below 18 years of the EU28, the seven environmental exposures (lead, PM_10_, PM_2.5_, ozone, secondhand smoke, dampness, and formaldehyde), were responsible for 210,777 disability-adjusted life years (DALYs) annually (Table 2). Fifty-nine percent of these DALYs were attributable to particulate matter (PM_10_ and PM_2.5_) exposure, 20% to secondhand smoke, 11% to ozone, 6% to dampness, 3% to lead, and 0.2% related to formaldehyde (Figure 1).

#### 3.1.1. Particulate Matter less than 10 Micrometer of Diameter (PM_10_) and less than 2.5 Micrometer of Diameter (PM_2.5_)

PM_10_ is associated with infant mortality (<1 year old) [8] and asthma (5–18 years old) [8]. Of these, infant mortality was associated with the largest burden (93,147 DALYs annually), followed by asthma (13,904 DALYs annually) (Table 2). PM_2.5_ is associated with low respiratory infections (<18 years old) [9] and was estimated to produce 17,453 DALYs annually.

#### 3.1.2. Secondhand Smoke

Secondhand smoke is associated with asthma (<14 years old) [13], low respiratory infections (<5 years old) [2], and otitis media (<5 years old) [2]. Of these, asthma was the disease with the largest burden, resulting in 20,880 DALYs annually, followed by low respiratory infections with 9728 DALYs annually, and Otitis media with 2062 DALYs annually (Table 2).

#### 3.1.3. Ozone

Ozone is associated with low respiratory symptoms (including cough) (5–14 years old) [10]. The number of cough days is related with ozone and was estimated to result in 10,057 DALYs annually. The number of days with low respiratory symptoms is related to ozone and was estimated to result in 14,122 DALYs annually.

#### 3.1.4. Dampness

Dampness is associated with asthma in children less than 14 years old [12] and was estimated to result in 12,954 DALYs annually.

#### 3.1.5. Lead

Lead exposure is associated with mild mental retardation (<14 years old) [2,14] and was estimated to result in 6216 DALYs annually.

#### 3.1.6. Formaldehyde

Formaldehyde is associated with asthma in children less than 3 years old [11] and was estimated to result in 33 DALYs annually.

#### 3.1.7. Sensitivity Analysis

Sensitivity analysis for PM_10_ and asthma assuming a counterfactual exposure of 1.9 μg/m^3^ resulted in 18,681 DALYs, and assuming a counterfactual exposure of 20 μg/m^3^ resulted in 3885 DALYs. We also performed a sensitivity analysis using a different exposure-response function between PM_10_ and asthma from a new meta-analysis; in this analysis, we estimated 45,098 DALYs [24]. The sensitivity analysis of PM_10_ and infant mortality assuming a counterfactual exposure of 1.9 μg/m^3^ resulted in 124,794 DALYs, and assuming counterfactual exposure of 20 μg/m^3^ resulted in 30,499 DALYs (see Appendix A). Sensitivity analysis for secondhand smoke assuming the minimum percentage of secondhand smoke reported in the European Union 28, resulted in 12,848 DALYs for asthma, 6867 DALYs for low respiratory infections, and 1168 DALYs for otitis media (see Appendix A). Sensitivity analysis for dampness and asthma, using mold (instead of dampness) as an exposure resulted in 11,470 DALYs (see Appendix A). Finally, the sensitivity analysis for formaldehyde and asthma using 60 μg/m^3^ as a threshold resulted in 1667 DALYs (see Appendix A).

## 4. Discussion

Our results show a large impact of environmental exposures on child health across Europe. This study found that the environmental risk factor for child health in the EU28 with the largest impact was air pollution (PM_10_, PM_2.5_, and ozone) exposure, representing more than two thirds of the environmental burden of disease of the seven exposures combined. This is similar to a previous burden of disease study in six countries of Europe [3]. Secondhand smoke also showed a large impact, resulting in 20% of the environmental burden of disease in European children.

In our analysis, we included three main air pollutants, PM_10_, PM_2.5_, and ozone. Ozone and particulate matter have shown independent health effects. Different health outcomes were chosen for PM_10_ and PM_2.5_. Specifically for particulate matter, scientific evidence suggests multiple health outcomes, such as respiratory, cardiovascular, neurological, or metabolic diseases, among others [25,26], but it was not possible to include all these outcomes in our analysis, mainly because of the lack of robust evidence (as described in the methods section) for children. The health outcomes in children that were included in this analysis for PM_10_ were asthma and infant mortality [8,27], and for PM_2.5_ we included lower respiratory infections [9] (see Table 1). Besides particles, we included in our analysis another major pollutant, ozone, that has been shown to have clear and independent impacts in children [10]. Ozone has been associated with impacts on the respiratory system, especially with asthma [28], and respiratory symptoms (such as cough and other low respiratory symptoms) [10]. In our analysis we included cough days and days with other low respiratory symptoms as the main outcomes to ozone exposure in children, as suggested by previous assessments [10]. Nitrogen dioxide (NO_2_) has not been included in this assessment to avoid double counting, since NO_2_ has a strong correlation with particulate matter [27].

Overall, air pollution (PM_10_, PM_2.5_, and ozone) has the largest health impact in our analysis (70% of all DALYs). From the 28 countries included in this analysis, 22 countries (with the exception of Luxemburg, Ireland, Sweden, Estonia, Finland, and Denmark) have reported levels of PM_10_ above the recommendations of the WHO (annual mean less than 20 μg/m^3^ of PM_10_) [29]. Ozone levels above the level recommended by the WHO (100 μg/m^3^ 8-hour mean) [29] have been reported in all the EU28. Our analysis for ozone assumed as counterfactual exposure a uniform distribution in the range of 33.3–41.9 ppb, for PM_10_ it assumed a counterfactual exposure of 7.9 μg/m^3^, and for PM_2.5_ a uniform distribution in the range of 2.4–5.9 μg/m^3^ [2,30]. Recent evidence has suggested that the counterfactual could be lower (1.4 μg/m^3^ for PM_2.5_, corresponding to 1.9 μg/m^3^ of PM_10_) [31], so we conducted a sensitivity analysis using these alternative counterfactual exposure concentrations, resulting in an increase of 34% of the impacts (18,681 DALYs) associated with PM_10_ (see Appendix A). A second sensitivity analysis was performed, using as a counterfactual exposure the recommended concentrations from the WHO guidelines of air quality for PM_10_ (annual mean less than 20 μg/m^3^) [29]. This sensitivity analysis shows what the benefits in child health would be (3885 DALYs) if the recommended WHO levels were reached in the EU28 (see Appendix A). These results highlight the importance of implementing specific measures to reduce child air pollution exposure in Europe.

The main sources of air pollution in Europe are well known and include agriculture, traffic, heat, power generation, and industry [32]. Since most of the European population lives in urban areas, exposure to pollution from road traffic is the main source [33]. Therefore, specific policies across Europe aimed to reduce traffic emissions should be implemented. Recent studies have suggested a list of effective interventions to reduce air pollution in the European context through low emission zones [34], a shift to walking, cycling and public transport [35], diversification of land use and increase of the population density in cities [36], and improvements in agricultural practices [37], among others.

Secondhand smoke had the second largest impact (20% of all DALYs) in our environmental burden of disease assessment. Like air pollution, secondhand smoke has been linked to multiple health outcomes [38,39] but few dose-response functions are available to be used in a risk assessment for children [3]. In our analysis, we included three secondhand smoke related diseases, asthma [13], lower respiratory infections [2], and otitis media [2]. Of those, asthma had the largest impact on DALYs, followed by low respiratory infections and otitis media. The percentage of secondhand smoke exposure among children less than 14 years old in the EU28 ranged between 15% (in Finland) and 54% (in Cyprus) [2]. We performed a sensitivity analysis applying the lowest percentage of secondhand smoke exposure in the 28 European countries (see Appendix A). This analysis showed the impact of a simple achievable and realistic goal for all the countries. If the 28 European countries could reduce secondhand smoke to 15% (as in Finland), 20,883 DALYs annually related to child health could be avoided. These benefits could be achieved by applying individual and population-based measures to reduce tobacco use in adults. Individual measures that have been described as effective smoking cessation interventions are behavior therapy, pharmacological therapy (e.g., bupropion), physician advice, nicotine replacement therapy, individual and telephone counseling, nursing, and self-help interventions [40]. Population-based interventions that have also shown efficacy are smoking bans in public spaces [41] and antismoking media campaigns [42].

Dampness has been associated with multiple respiratory outcomes like upper respiratory tract symptoms, cough, wheeze, and asthma [43]. We included asthma as the main health outcome related to dampness because it has robust evidence [12]. The proportion of houses with dampness was used as a measure of exposure and ranged in the EU28 between 5% in Malta and 30% in Slovenia. Scientific evidence has suggested a causal pathway between dampness, mold and asthma. In our analysis, we included only dampness as exposure, but we also ran a sensitivity analysis assuming mold as main exposure (see Appendix A). In this sensitivity analysis we estimated similar results between dampness and mold (12,954 DALYs annually using dampness vs. 11,470 DALYs annually using mold). A review on housing interventions to control asthma related to mold and dampness has suggested combined elimination of moisture intrusion and leaks and removal of moldy items for the reduction of health risks [44].

Lead exposure in children has been widely studied and has been particularly associated with neurodevelopmental outcomes [5]. In our assessment, we considered the impact of blood lead levels on loss of IQ points and their translation to mild mental retardation [2,14,45], and it was estimated that lead contributes 3% to the environmental DALYs. The blood lead levels reported in the EU of 28 countries ranged between 13 μg/l in Sweden to 104 μg/l in Romania [21]. Lead in the environment has multiple sources, including industrial processes, paint, petrol, solder in canned foods, and water pipes [21]. It has been suggested that lead in atmospheric air or flaked paint is deposited in soil and dust and could be ingested by children, increasing their blood lead levels [46]. In addition, food and water may also be important media of baseline exposure to lead [46]. Public health measures to reduce and prevent exposure to lead have also been widely proposed [45,46]. Some of these measures are phasing out lead additives in fuels and removing lead from petrol, reducing and phasing out the use of lead-based paints, eliminating the use of lead in food containers, traditional medicines and cosmetics, and minimizing the dissolving of lead in water treatment and water distribution systems.

Formaldehyde is a chemical used in industrial processes, building materials, and in a wide range of products [47]. Formaldehyde is widely prevalent both indoors and outdoors, but it reaches high levels mostly indoors. Formaldehyde has been related in adults to nasal cancer, leukemia, and respiratory outcomes [47] and in children, it has been related to asthma [11]. In our environmental burden of disease assessment, formaldehyde results in a small impact on children (less than 0.2% of all DALYs). In Europe levels are mostly under the guideline value (100μg/m^3^) [47]. We performed a sensitivity analysis using an alternative threshold (60μg/m^3^) as suggested by previous assessments [48] (see Appendix A). In this sensitivity analysis we found a larger environmental burden of disease associated with formaldehyde (33 DALYs annually in the main analysis vs 1667 DALYs annually).

As in all burden of disease assessments, our study was limited by the availability of data and the necessity to make assumptions to model likely scenarios. In particular, the lack of data and the variability in availability across the EU28 was the main limitation in our study. Another limitation in the burden of disease approach, is the lack of robust evidence of exposure-response relationships between pollutants and health outcomes. Previous burden of disease approaches have suggested placing greater value upon years lived by younger populations than older populations. We chose not to include any age discount or adjust for this in our analysis. This could be a limitation because the non-discounting approach could result in less conservative results. Another limitation is the use of disability weights (DW) and disease duration (L) from the general population and applying these to children. This was done due to the lack of a specific DW and L for children. The inclusion criteria used in this assessment for pollutants and outcomes reduce the possibility of including more pollutants or outcomes which have been studied in children. Chemical pollutants, such as bisphenol, or physical pollutants, such as radiation, with high social interest were not included in this study due to the lack of data availability across the EU28 and/or the lack of strength of evidence to quantify the impact on a specific health outcome. Including only those risk factors with a strong causal relationship, supported by large cohort studies and meta-analyses, is a strength of this assessment, ensuring a more robust and conservative approach [49]. The conservative estimation of this environmental burden of disease analysis considers only seven environmental risks for children in Europe. Stakeholders should use this study to promote actions aimed to reduce those seven environmental risk factors. In addition, stakeholders should have in mind other relevant environmental hazards (not included in this study), such as plastics, pesticides, ionizing and non-ionizing radiations, noise or climate change, among others, that have less available information across Europe and require research support focused on children and promotion of mitigation actions.

To reduce the uncertainty of our analysis, we included confidence intervals for all estimates made. We also ran several sensitivity analyses to identify the main input data that could impact more in the magnitude of our estimates (see Appendix A). This is the first environmental burden of disease study focused on the EU28 that included seven environmental risk factors focused on children. Previous assessments have only included a few European countries and focused only on one or a few exposures [7], or not focused particularly on children [3]. Trasande et al. [7] have produced a specific assessment of endocrine disruptors in the EU28. This specific group of pollutants has not been included in our study because we decided to focus only on those environmental exposures that have shown associations based on meta-analysis and not only on expert judgment.

The assessment of environmental burden of disease estimates offers the opportunity to identify priorities and solutions for policy and research. These priorities and solutions are presented in this study as recommendations for authorities, public health specialist, and researchers (Table 3) and are as follows. First, this study found a real need to create and implement effective policies to reduce child exposure to environmental risk factors across Europe, with special attention to major risk factors such as air pollution and secondhand smoke. Second, there is a need to create common European databases, which collect and harmonize exposure data for (old and new) environmental risk factors, especially for children. Third, there is a need for epidemiological studies on multiple environmental risk factors, with special attention to providing dose-response functions with harmonized exposure and outcome definitions. Currently, a number of European research projects focus on implementing the “exposome” concept (HELIX [50]), EXPOsOMICs [51] and HEALS [52], and some of them are collecting relevant information on multiple exposures and outcomes in children. These projects are a good example to start filling some of these gaps.

## 5. Conclusions

Environmental risk factors for children across Europe impact public health significantly. Air pollution and secondhand smoke provide the biggest burden in children across Europe. Evidence-based policies to reduce child exposure to environmental risk factors are urgently needed in Europe. More research to collect and harmonize data between a wider range of environmental exposures and health outcomes in children is needed.

## Figures and Tables

**Figure 1 ijerph-16-01084-f001:**
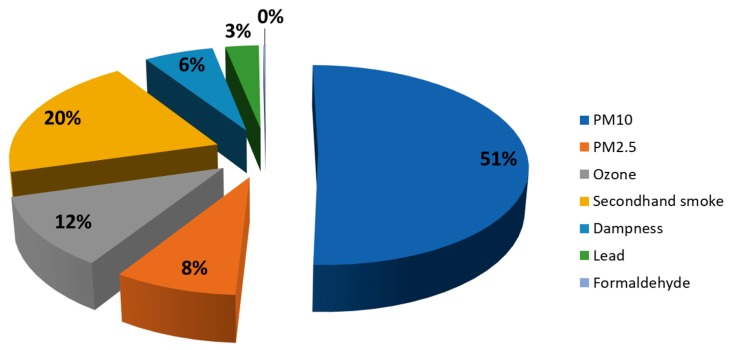
Percentage of exposures studied that contribute to the environmental burden of disease in EU28, in the population less than 18 years old. PM_10_: particulate matter less than 10 micrometers of diameter; PM_2.5_: particulate matter less than 25 micrometers of diameter.

**Table 1 ijerph-16-01084-t001:** Exposures, health outcomes and dose response functions included in the analysis.

Risk Factor	Health Outcome	Population	Exposure Estimate	Unit of Exposure	DRF	Reference
PM_10_	Asthma	5–18 years	Ambient levels	μg/m^3^	1.028 (1.0006–1.051)	[8]
PM_10_	Infant mortality	<1 year	Ambient levels	μg/m^3^	1.04 (1.02–1.07)	[8]
PM_2.5_	Low respiratory infections	<18 years	Ambient levels	μg/m^3^	Function *	[9]
Ozone	Cough days	5–14 years	Ambient levels	μg/m^3^	0.093 (0.019–0.22)	[10]
Ozone	Low respiratory symptoms	5–14 years	Ambient levels	μg/m^3^	0.016 (−0.043–0.08)	[10]
Formaldehyde	Asthma	<3 years	Indoor levels	μg/m^3^	1.017 (1.004–1.025)	[11]
Dampness	Asthma	<14 years	Percent exposed	House Yes/no	1.33 (1.12–1.56)	[12]
Mold	Asthma	<14 years	Percent exposed	House Yes/no	1.29 (1.04–1.6)	[12]
SHS	Asthma	<14 years	Percent exposed	Parental Yes/no	1.32 (1.24–1.41)	[13]
SHS	Low respiratory infections	<5 years	Percent exposed	Parental Yes/no	1.55 (1.42–1.69)	[2]
SHS	Otitis media	<5 years	Percent exposed	Parental Yes/no	1.37 (1.24–1.50)	[2]
Lead	IQ loss	<5 years	Blood levels	mg/L	0.051 (0.032–0.07)	[14]
Lead	Mild mental retardation	<5 years	Blood levels	mg/L	Function *	[14]

PM_10_: particulate matter less than 10 micrometers of diameter; PM_2.5_: particulate matter less than 2.5 micrometers of diameter; SHS: second hand smoke; IQ: intelligence quotient. DRF: dose response function. * These DRF imply an equation or multiple step equation. Please consult Burnett [9] for low respiratory infections and Landphear et al. [14] for mild mental retardation.

**Table 2 ijerph-16-01084-t002:** Environmental burden of disease in EU28, in the population less than 18 years old.

Risk Factor	Health Outcome	Population	Cases	DALYs	LCI	UCI	DALYs/100,000 Population	Total DALYs by Exposure
Lead	Mild mental retardation	<5 years	138,646	6216	2699	11,414	15.09	6216
PM_10_	Asthma	5–18 years	43,402	13,904	462	15,181	17.80	
	Infant mortality	<1 year	1078	93,147	45,106	166,668	16,324	107,051
PM_2.5_	Low respiratory infections	<18 years	134,032	17,453	8042	29,659	21.14	17,453
Secondhand smoke	Asthma	<14 years	106,085	20,880	15,645	25,065	28.03	
	Low respiratory infections	<5 years	142,530	9728	5942	14,040	37.79	
	Otitis media	<5 years	821,499	2062	1132	3396	8.01	42,501
Ozone	Cough days	5–14 years	52,436,762 *	10,057	902	10,177	19.78	
	Low respiratory symptoms days	5–14 years	52,059,353 *	14,122	760	14,207	27.78	24,179
Dampness	Asthma	<14 years	65,815	12,954	3022	31,646	17.39	12,954
Formaldehyde	Asthma	<3 years	423	33	4	83	0.60	423
Total								210,777

* Days with cough or other low respiratory symptoms; PM_10_: particulate matter less than 10 micrometers of diameter; PM_2.5_: particulate matter less than 25 micrometers of diameter; DALY: disability-adjusted life years; LCI: lower confidence interval; UCI: upper confidence interval.

**Table 3 ijerph-16-01084-t003:** Recommendations for authorities, public health practitioners, and researchers.

Stakeholder	Recommendations
Authorities	Develop evidence-based policies to reduce child exposure to environmental risk factors across Europe, with special attention to major risk factors as secondhand smoke and air pollution
Public health practitioners	Create and develop European health databases with harmonized exposure data for (old and new) environmental risk factors for children across all European countries
Researchers	Develop epidemiological studies on multiple environmental risk factors, with special attention to providing dose-response functions with harmonized exposure and outcome definitions

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
