# Peer review of "Environmental Burden of Childhood Disease in Europe"

_ijerph, 2019, doi:10.3390/ijerph16061084_

Reviewer 1 Report

This manuscript to assess the burden of childhood disease due to environmental risks across the European Union. Scientific evidence shows clearly that environmental risk factors affect human health. Environmental exposures are known to be important contributors to the global burden of disease among children and adolescents, but there are still gaps in our knowledge about the magnitude and regional distribution of the environmental burden in the childhood, what makes it an important topic of research.

At the end of the Section 1, I suggest to authors insert a paragraph with structure of manuscript.

At the Section 5, I suggest to authors insert the limitations of work done.

Author Response

Response to Reviewer 1 

Dear reviewer, we thanks your valuable comments, please find below answer to your suggestions.

Reviewer: At the end of the Section 1, I suggest to authors insert a paragraph with structure of manuscript.

Answer: 

Dear reviewer, we have added this paragraph with the structure of the manuscript, see below:

“This document was divided into five sections (introduction, methods, results, discussion, and conclusion). Methods section describe the exposures and outcome selection, data collection and burden of disease methods. Results section summarized the impacts on EU of 28 counties and described the results by each exposure and the sensitivity analysis. Discussion section compares the results with previous studies and explain the differences between exposures and geographical distributions, and includes a list of recommendations for authorities, public health practitioners and researchers.”

Reviewer: At the Section 5, I suggest to authors insert the limitations of work done.

Answer: Dear reviewer we modified the discussion section adding a better explanation related to the limitations in our study:

“As in all burden of disease assessments, our study was limited by the availability of data and the necessity to make assumptions to model likely scenarios. In particular, the lack of data and the variability in availability across the EU28 was the main limitation in our study. Another limitation, in the burden of disease approach, is the lack of robust evidence of exposure-response relationships between pollutants and health outcomes. The previous burden of disease approaches has suggested to considered more valuable the years lived by a younger population than older populations. We chose not to include any age discount or adjust in our analysis. This could be a limitation because non-discounting approach could result in less conservative results. Another limitation is the use of disability weights (DW) and disease duration (L) from the general population and applied these for children. These have been done due to the lack of a specific DW and L for children. The inclusion criteria used in this assessment for pollutants and outcomes, reduce the possibility to include more pollutants or outcomes which have been studied in children. Chemical pollutants, such as bisphenol, or physical pollutants, such as radiation, with high social interest, where not included in this study due to the lack of data availability between the EU28, and/or the lack of strength of evidence to quantify the impact on a specific health outcome. Include only those risk factors with a strong causal relationship, supported by large cohorts studies and meta-analysis is a strength of this assessment, by providing the more robust, and at the same time a conservative approach as suggested to be in all risk assessments (Murray et al. 2003). The conservative estimation in this environmental burden of disease considers only seven environmental risks for children in Europe. Stakeholders should use this study to promote actions aimed to reduce those seven environmental risk factors. Also, stakeholders should have in mind other relevant environmental hazards (not included in this study), such as plastics, pesticides, ionizing and non-ionizing radiations, noise or climate change, between others, that have less available information across Europe, and require research support focused on children and prompt mitigation actions.”

Reviewer 2 Report

This paper is in its failure to address major environmental causes of health in children. It focuses on seven environmental risk factors, none of which can be blamed on industry or environmental contamination. It totally ignores a number of sources which have a significant literature associated with them, which the authors of this contribution cannot fail to have known. There are at least three elephants in this room. The two main ones are ionizing radiation exposure through radioactive contamination which has caused significant genomic damage in Europe and increasingly EM radiation from mobile phones, their radiation masts and other devices. Even the WHO now classify mobile phone radiation as a possible carcinogen and two recent major studies, in the USA and Italy have found a wide range of cancer effects in rats. The issue of childhood cancer is not even mentioned in the paper even though almost every study of childens’ cancer near nuclear sites shows statistically significant effects. There are also radon effects. Additionally we see no mention of chemical contamination from landfill or waste streams (heavy metals, PAH, Uranium, other carcinogens). There are also the immediate and downstream side effects of multiple vaccinations, found in a number of unpublished studies to be highly significant.

As the abstract states clearly, its aim is to advise politicians and health authorities about what they should be focusing on. With the one exception of lead, the paper is a manifesto for the environmental contamination industry and the nuclear industry. There are more than 20 studies showing the increases in congenital malformations in European and other countries after Chernobyl.  A recent review by Schmitz Feuerhake et al ( 2006) is cited below. This also seems to have passed David et al by.

The process lasted several years and resulted in a number of publications, including one on ionizing radiation and child health. I attach the references below. David et al seem to have missed these.

I guess they could totally re-write it and mention the selection they made, also mention the other stuff, and entitle it, “Some selected contributors to the burden of environmental disease in children in Europe”.

Useful references for David et al

1. Schmitz-Feuerhake, Busby C, Pflugbeil P  Genetic Radiation Risks-A Neglected Topic in the Low Dose Debate. Environmental Health and Toxicology.  2016. 31Article ID e2016001. http://dx.doi.org/10.5620/eht.e2016001.

2. Busby C and Fucic A (2006) Ionizing Radiation and children’s health: PINCHE conclusions Acta Paediatrica S 453  81-86

3. Van den Hazel P, Zuurbier M, Bistrup M L, Busby C, Fucic A, Koppe JG et al (2006) Policy and science in children’s health and environment: Recommendations from the PINCHE project. Acta Paediatrica S 453 114-119

4. Koppe JG, Bartonova A, Bolte G, Bistrup ML, Busby C, Butter M et al (2006) Exposure to multiple environmental agents and their effects. Acta Paediatrica S 453 106-114

5. Van den Hazel P, Zuurbier M, Babisch W, Bartonova A, Bistrup M-L, Bolte G, Busby C et al, (2006) ‘Today’s epidemics in children: possible relations to environmental pollution’ Acta Paediatrica S 453 18-26

Author Response

Dear reviewer, we thank your comments and vision for improving our manuscript. We agreed that many environmental risk factors are not included in the scope of this paper. We have included some sentences in the document to highlight the importance of other risk factors that are not involved in the document, in terms of mitigation actions and research. We also clarify the reason for not include more risk factors, mainly because the lack of harmonizing data between the EU28 and/or lack of large cohort studies or meta-analysis in humans, specifically in children to quantify such impacts. Please find below some sentences that we have added to acknowledge your comments.

In the introduction section we have added:

“These seven environmental risk factors were chosen, between all the possible environmental risk factors for children, based on the strength of evidence on the causal relationship between the risk factor and a health outcome, and the data (populational, health and exposure) availability across the EU of 28 countries. The environmental risk factors were defined for this study as a physical or chemical environmental exposure that harms the health of the children. The scope of this study does not include non-environmental risk factors, such as lifestyle, metabolic, genetic, or those related to access to health services.”

In the discussion section we have added:

“Chemical pollutants, such as bisphenol, or physical pollutants, such as radiation, with high social interest, where not included in this study due to the lack of data availability between the EU28, and/or the lack of strength of evidence to quantify the impact on a specific health outcome.”

“The conservative estimation in this environmental burden of disease considers only seven environmental risks for children in Europe. Stakeholders should use this study to promote actions aimed to reduce those seven environmental risk factors. Also, stakeholders should have in mind other relevant environmental hazards (not included in this study), such as plastics, pesticides, ionizing and non-ionizing radiations, noise or climate change, between others, that have less available information across Europe, and require research support focused on children and prompt mitigation actions.”

Reviewer 3 Report

This is a well written manuscript which estimates the burden of childhood disease due to environmental risk factors in the European Union.  The authors have done a great job and I find it difficult to fault the paper.

My only suggestions are that they could add to the abstract that over half the annual DALYs were from infant mortality due to PM10.

Equation 3, line 103, page 3 contains a typo.

Author Response

Response to Reviewer 3

Dear reviewer, we thanks your valuable comments, please find below answer to your suggestions.

This is a well written manuscript which estimates the burden of childhood disease due to environmental risk factors in the European Union.  The authors have done a great job, and I find it difficult to fault the paper.

Reviewer: My only suggestions are that they could add to the abstract that over half the annual DALYs were from infant mortality due to PM10.

Answer:

Thanks for your comment, we have included this sentence in the abstract:

“Among the included environmental risk factors, air pollution (particulate matter and ozone) was estimated to produce the highest burden of disease in children in Europe, with half of which was specifically related to PM10 infant mortality.”

Reviewer: Equation 3, line 103, page 3 contains a typo.

Answer: Thanks for appointing this typo, we have corrected the line:

 “(3) DALY = AP x DW x L”

Reviewer 4 Report

REVIEW IJERPH- 2019-16

The authors have conducted a HIA of different known environmental stressors in Europe. It is an important and well-written paper. I have some concerns listed below.

1. Line 52-53 Please explain what you mean with expert judgment and not robust evidence.

2. For readability, it is good to keep method section short and concise but here I find some lacking information. Please elaborate on exposure data a bit more.

3. Line 86 is PM10 missing from this sentence? (in table 1 you write PM10)

4.  In HRAPIE, the experts concluded that Europe-wide modelling for particles is only available for PM2.5; so, in cases where health effects are expressed against PM10 in the literature, a conversion has to be employed to assess the equivalent impact per unit of PM2.5 often  assuming a PM2.5/PM10 ratio of 0.65 [1]. Did you use PM10 models or a conversion rate? In the case of a conversion rate, was the same rate used for all countries?

5.  For which year did you do the HIA? Was it same year for all data collection?

6.  Counterfactual level should be stated in method.

7.  Counterfactual levels will heavily affect results. Hänninen et al. used no threshold and prior studies have used different levels. Review on this and other choices can be found here https://www.ncbi.nlm.nih.gov/pubmed/29404862 The choices you made are found in discussions and should maybe be in method?

8.  Please indicate further on method used for discounting. There is an ongoing discussion whether future years of healthy life are valued less than those at present time (discounting) and if a year lived as a young adult is worth more than a year lived at younger or older ages. This can be accounted for by discounting or age weighting the results and is optional when calculating DALYs. For example, the EBD by Hänninen et al. has chosen not to discount or age weight their estimates as it often makes health impacts in children weigh less with the argument that this would not be in line with the priorities set in the European Health Action Impact plan (WHO 2010b) that made children’s health a priority.  This should at least be mentioned in discussion.

9.  With increasing evidence, HRAPIE opened up for using NO2 but to avoid double counting, would using this estimate influence your results?  

10. Please see https://insights.ovid.com/crossref?an=00001648-200711000-00021 for more detailed discussion of on using DALY and recommendation for the use of terms such as ‘average loss of life expectancy’.

11. The authors present discuss results in Discussion that have not been presented in results. Consider this.

12. Noise and climate change impact such as heat was not included and it should maybe be mentioned why as their impact might be larger than the ones you mentioned.

13. Table 3, lead is mentioned but not the one with larger impact.

14. Sentence "This decision …." on line 289-291 is a bit unclear, consider rephrasing.  

Author Response

Response to Reviewer 4

Dear reviewer, we thanks your valuable comments, please find below answer to your suggestions.

Reviewer: 1. Line 52-53 Please explain what you mean with expert judgment and not robust evidence.

Answer:

Thanks for your comment, we have included these explanations in the introduction section:

“These studies have focused on specific group of environmental pollutants (Trasande et al. 2015), or based their estimations on expert judgment (using qualitative approaches, without including a comparative risk assessment) and not on robust evidence (using studies other than large cohort studies or meta-analysis) (Bartlett and Trasande 2014; Landrigan et al. 2002; Trasande et al. 2015).”

Reviewer: 2. For readability, it is good to keep method section short and concise but here I find some lacking information. Please elaborate on exposure data a bit more.

Answer:

Thanks for your comment, we have added more detail on the exposure data in the methods section:

Lead exposure data from IHME (Institute for Health Metrics and Evaluation 2016b) database was primarily extracted from the literature regarding blood lead, in addition to a blood lead surveys. Blood lead values were derived from studies that take blood samples and analyze them using various techniques to determine the level of lead present. The theoretical minimum-risk exposure level (TMREL) for lead used was 2.0 mg/dL. For PM10 and PM2.5 exposure estimates, came from the IHME (Institute for Health Metrics and Evaluation 2016b), was drawn from multiple sources, including satellite observations of aerosols in the atmosphere, ground measurements, chemical transport model simulations, population estimates, and land-use data. Monitor-specific measurement rather than city averages was used. The measurements were recorded in 2014. For locations measuring only PM10, PM2.5 measurements were estimated from PM10. his was performed using a locally derived conversion factor (PM2.5/PM10 ratio, for stations where measurements are available for the same year) that was estimated using population-weighted averages of location-specific conversion factors for the country. Satellite estimates were available at 11 x 11 km resolution and combine aerosol optical depth retrievals from multiple satellites with the GEOS Chem chemical transport model and land use information. Estimates of the sum of particulate sulfate, nitrate, ammonium, and organic carbon and the compositional concentrations of mineral dust simulated using the GEOS Chem chemical transport model, and a measure combining elevation and the distance to the nearest urban land surface were available for 2000 to 2015 for each 11 x 11 km grid cell. The TMREL for PM10 was assumed 7.9 g/m3, and uniform distribution from 2.4 –5.9 g/m3 of PM2.5. For ozone, exposure data were derived from the TM5-FASST chemical transport model, which generates a 3-month running average of daily 1-hour maximum ozone values at the 0.1°×0.1° for the years 1990, 2000, and 2010 (Institute for Health Metrics and Evaluation 2016b). The TMREL of ozone was also defined based on the exposure distribution from American Cancer Society CPSII study. A uniform distribution was drawn around the minimum and 5th percentile values experienced by the cohort. This value was 33.3 - 41.9 ppb. Secondhand smoke exposure was estimated based on the Health Behavior in School-aged Children survey collected between 2013 and 2014. The TMREL for secondhand smoke was zero exposure. For dampness, exposure levels where derived from the Eurostat Statistics for Income and Living Conditions (SILC) data (Institute for Health Metrics and Evaluation 2016b). The TMREL for dampness was zero exposure. For formaldehyde, country levels where obtained from national indoor concentration data reported in the scientific literature (Hänninen et al. 2014). For those countries without national concentration report, the lowest level reported between other European countries was used. The TMREL level used for formaldehyde was 100 mg/m3.”

Reviewer: 3. Line 86 is PM10 missing from this sentence? (in table 1 you write PM10)

Answer:

Thanks for highlit this, we have added PM10 in the sentence.

“Exposure data were collected from IHME (Institute for Health Metrics and Evaluation 2016b) for lead,PM10, PM2.5, ozone and secondhand smoke, and Environmental and Health Information System (ENHIS)(WHO/Europe 2012) for dampness (see supplemental material).”

Reviewer: 4. In HRAPIE, the experts concluded that Europe-wide modelling for particles is only available for PM2.5; so, in cases where health effects are expressed against PM10 in the literature, a conversion has to be employed to assess the equivalent impact per unit of PM2.5 often  assuming a PM2.5/PM10 ratio of 0.65 [1]. Did you use PM10 models or a conversion rate? In the case of a conversion rate, was the same rate used for all countries?

Answer:

Thanks for your comment we have included a description of the PM2.5/PM10 ratios used for each country in the methods section:

“For locations measuring only PM10, PM2.5 measurements were estimated from PM10. This was performed using a locally derived conversion factor (PM2.5/PM10 ratio, for stations where measurements are available for the same year) that was estimated using population-weighted averages of location-specific conversion factors for the country.”

Reviewer: 5. For which year did you do the HIA? Was it same year for all data collection?

Answer:

The HIA was done using for the year 2015. The year population data was 2015 and the exposure data used was reported between 2013- 2015). We have included this description in the methods section:

“The environmental burden of childhood disease was estimated for the year 2015, using population data from 2015 and exposure date ranging from 2013 to 2015.”

Reviewer: 6. Counterfactual level should be stated in method.

Answer:

Thanks for your suggestion, we have included the theoretical minimum-risk exposure level for each other exposures in the methods section. Please see the answer above.

Reviewer: 7. Counterfactual levels will heavily affect results. Hänninen et al. used no threshold and prior studies have used different levels. Review on this and other choices can be found here https://www.ncbi.nlm.nih.gov/pubmed/29404862 The choices you made are found in discussions and should maybe be in method?

Answer:

Thanks for your comment, we agree with the reviewer on the relevance of choosing the counterfactual levels. We have included the theoretical minimum-risk exposure level for each other exposures in the methods section. Please see the answer above.

Reviewer: 8. Please indicate further on method used for discounting. There is an ongoing discussion whether future years of healthy life are valued less than those at present time (discounting) and if a year lived as a young adult is worth more than a year lived at younger or older ages. This can be accounted for by discounting or age weighting the results and is optional when calculating DALYs. For example, the EBD by Hänninen et al. has chosen not to discount or age weight their estimates as it often makes health impacts in children weigh less with the argument that this would not be in line with the priorities set in the European Health Action Impact plan (WHO 2010b) that made children’s health a priority.  This should at least be mentioned in discussion.

Answer:

We have chosen not to apply any discounting or age weight in our analysis, as Hanninen et al. We have highlighted this in the discussion section.

“The previous burden of disease approaches has suggested to considered more valuable the years lived by a younger population than older populations. We chose not to include any age discount or adjust in our analysis.”

Reviewer: 9. With increasing evidence, HRAPIE opened up for using NO2 but to avoid double counting, would using this estimate influence your results?  

Answer:

Thanks for the comment, we agree that include NO2, accounting for possible double counting could modify our results. We are currently assessing the impacts of NO2 using new NO2 land use regression model for Europe, but unfortunately, we were not able to include this analysis yet in this manuscript.

Reviewer: 10. Please see https://insights.ovid.com/crossref?an=00001648-200711000-00021 for more detailed discussion of on using DALY and recommendation for the use of terms such as ‘average loss of life expectancy’.

Answer:

Thanks for this valuable reference, we agreed with the authors with the idea to provide multiple health outcome values and avoid any misleadings in terms of deaths avoided, and preferred to use postponed. We have reviewed our manuscript to identify any missleading messages with this terminology.

Reviewer: 11. The authors present discuss results in Discussion that have not been presented in results. Consider this.

Answer:

Thanks for this comment, we have added more information in the methods sections that previously have been only presented in the discussion. Please see above (answer to question 2)

Reviewer: 12. Noise and climate change impact such as heat was not included and it should maybe be mentioned why as their impact might be larger than the ones you mentioned.

Answer:

Thanks for your comment, we agreed with the reviewer on the relevance of other environmental risk factors, such as noise or climate change, and their impact on populational health. We have highlighted these risk factors and suggested to encourage more support for research on those risk factor in children, plus mitigation actions.

We have included this sentence in the discussion:

“Also, stakeholders should have in mind other relevant environmental hazards (not included in this study), such as plastics, pesticides, ionizing and non-ionizing radiations, noise or climate change, between others, that have less available information across Europe, and require research support focused on children and prompt mitigation actions.”

Reviewer: 13.   Table 3, lead is mentioned but not the one with larger impact.

Answer:

Thanks for your comment. We have updated table 3 and substitute lead by secondhand smoke.

Reviewer: 14.   Sentence This decision ….on line 289-291 is a bit unclear, consider rephrasing.  

Answer:

Thanks for your comment, we have modified the sentence, to include more clarity.

“Include only those risk factors with a strong causal relationship, supported by large cohorts studies and meta-analysis is a strength of this assessment, by providing the more robust, and at the same time a conservative approach as suggested to be in all risk assessments (Murray et al. 2003)”

Reviewer 5 Report

This is solid and well written manuscript describing a highly relevant study.  Its reliance on population representative exposure data, quantitative human dose/response data, and burden of disease modelling allow the different risks and the achievable benefits to be quantified and compared, which is essential for informed risk management decisions.  I have no general critical comments about it, and I recommend it to be published without delay. There are, however, some minor comments and a few points that need clarification or a bit more opening up, and I have listed them below.

Page 2, Table 1. Check if the unit of exposure should be mg/m3 or μg/m3.  Also, only references are given to the Dose/Response functions for lower respiratory infections from PM2.5 exposure, and for mild mental retardation from lead exposure, but not the D/R functions themselves, in the table or in the text – amend.

Page 3, lines 100 – 114. The disability weights (DW) and estimates of duration (L) used have been collected and agreed for general population and these are not necessarily valid or relevant for children. Respective values for children are, to my knowledge, not available. Therefore I accept the use of general population values, but this issue should be pointed out and its possible consequences at least shortly treated in the DISCUSSION

Page 4, Table 2 and page 5, lines 132-138. The authors have treated the effect overlap issues of PM10 and PM2.5 by not including the same health outcomes for both. This is the correct way. However, in a similar way there is a highly significant exposure, and consequently health outcome overlap between the two PM metrics, i.e., PM2.5 forms in average about half of the PM10, and the proportion of the health effects assigned to PM10 that are actually due to PM2.5 [and not to PM(10-2.5) is poorly known, and can approach 100 %.  Consequently, it is Ok to sum the health effects assigned to PM10 and to PM2.5, but there are very limited data to support separation of the contributions from PM10 and PM2.5.

Line 138. The number 134 032 refers to cases, not DALYs. Correct to ’17 453 DALYs annually’.  

Page 5, line 161, to the end of DISCUSSION, correct several g/m3 to μg/m3

Page 6, line 196 – 198. Note that the double counting of NO2 and PM effects is not an issue because of the similarity of health outcomes [Re. asthma and low respiratory infections from both PM and from SHS] but instead because the PM and NO2 exposures are strongly correlated.

Line 216.  I would add …, land traffic, heat and power generation …  because local space heating is a major source of fine PM in most of Europe.

Page 7, lines 237-237. Just a suggestion: Unlike the other pollutants considered, SHS children’s exposure to SHS can be eliminated completely. It would therefore be of interest and relevance to also use SHS exposure prevalence of 0 % as another reference point.

Lines 267-271. For all I know, because of already implemented public health measures lead in petrol, lead-based paints, and lead in tin cans have been not only reduced, but outlawed within the EU countries decades ago?

Author Response

Response to Reviewer 5

Dear reviewer, we thanks your valuable comments, please find below answer to your suggestions.

Reviewer:  Page 2, Table 1. Check if the unit of exposure should be mg/m3 or μg/m3.  Also, only references are given to the Dose/Response functions for lower respiratory infections from PM2.5 exposure, and mild mental retardation from lead exposure, but not the D/R functions themselves, in the table or the text – amend.

Answer:

Thanks for your comment. We think the change into the journal format misleads the unit of exposures. We have reviewed the paper to correct all the unit of exposures in tables and text. We have also amended table 1 describing that  DRF from low respiratory infections and mild mental retardation are functions that are not directly presented in the table to be a specific function or multiple step function. But we have provided the specific references to consult each DRF in the table and as a footnote explanation.

“* These DRF, imply an equation or multiple step equation, please consult Burnett, 2014 for low respiratory infections and Landphear, et al 2005 for mild mental retardation.”

Reviewer:  Page 3, lines 100 – 114. The disability weights (DW) and estimates of duration (L) used have been collected and agreed for general population and these are not necessarily valid or relevant for children. Respective values for children are, to my knowledge, not available. Therefore I accept the use of general population values, but this issue should be pointed out and its possible consequences at least shortly treated in the DISCUSSION

Answer:

Thanks fro yoru comment, we agreed with the reviewer on this point, and we have added a sentence in the discussion section.

“Another limitation is the use of disability weights (DW) and disease duration (L) from the general population and applied these for children. These have been done due to the lack of a specific DW and L for children.”

Reviewer:  Page 4, Table 2 and page 5, lines 132-138. The authors have treated the effect overlap issues of PM10 and PM2.5 by not including the same health outcomes for both. This is the correct way. However, in a similar way there is a highly significant exposure, and consequently health outcome overlap between the two PM metrics, i.e., PM2.5 forms in average about half of the PM10, and the proportion of the health effects assigned to PM10 that are actually due to PM2.5 [and not to PM(10-2.5) is poorly known, and can approach 100 %.  Consequently, it is Ok to sum the health effects assigned to PM10 and to PM2.5, but there are very limited data to support separation of the contributions from PM10 and PM2.5.

Answer:

Thanks for your comment, we agree with the reviewer on the lack of evidnce to sum the contributions from PM10 and PM2.5. But as the reviewer also highlighted this is an accepted approach to deal with this assessment.

Reviewer: Line 138. The number 134 032 refers to cases, not DALYs. Correct to ’17 453 DALYs annually’.  

Answer:

Thanks for appointing this mistake, we have amended the text to reflect DALYs in this line.

Reviewer: Page 5, line 161, to the end of DISCUSSION, correct several g/m3 to μg/m3

Answer:

Thanks for appointing these typos, we think the translation to the journal format has prudence these typos. We have reviewed all the paper and fixed all the typos.

Reviewer:  Page 6, line 196 – 198. Note that the double counting of NO2 and PM effects is not an issue because of the similarity of health outcomes [Re. asthma and low respiratory infections from both PM and from SHS] but instead because the PM and NO2 exposures are strongly correlated.

Answer:

Thanks for this comment, we have amene the text including the reviewer comment.

“Nitrogen dioxide (NO2) has not been included in this assessment to avoid double counting since NO2 has a strong correlation with particulate matter.”

Reviewer:  Line 216.  I would add …, land traffic, heat and power generation …  because local space heating is a major source of fine PM in most of Europe.

Answer:

Thanks for this suggestion, we have added heat to the sentence

“The main sources of air pollution in Europe are well known, and include agriculture, land traffic, heat, power generation, and industry”

Reviewer:  Page 7, lines 237-237. Just a suggestion: Unlike the other pollutants considered, SHS children’s exposure to SHS can be eliminated completely. It would therefore be of interest and relevance to also use SHS exposure prevalence of 0 % as another reference point.

Answer:

Thanks for your comment, we agree with the reviewer and the main analysis take in consideration as a theoretical minimum-risk exposure level zero. We have added a better explanation in the methods section about this TMREL fro secondhand smoke.

“The TMREL for secondhand smoke was zero exposure.”

Reviewer: Lines 267-271. For all I know, because of already implemented public health measures lead in petrol, lead-based paints, and lead in tin cans have been not only reduced, but outlawed within the EU countries decades ago?

Answer:

The reviewer is right; these are the most known sources of exposure, and these have been tackled with multiple regulations in the past. But we wanted to aknowledge not only these as a good example for successful interventions if not other sources that still prevalent and more in terms of poor control or implementation of trade regulations in EU.
